# Towards a Sustainable Classroom Ecology: Translanguaging in English as a Medium of Instruction (EMI) in a Finance Course at an International School in Shanghai

**Xiaozhou (Emily) Zhou [1], Chenke Li [1,\*] and Xuesong (Andy) Gao [2]**

1   School of English Studies, Shanghai International Studies University, 550 West Dalian Road, Hongkou District, Shanghai 200083, China; xzhou@shisu.edu.cn
2   School of Education, University of New South Wales (UNSW Sydney), Sydney, NSW 2052, Australia; xuesong.gao@unsw.edu.au
\*   Correspondence: 0171101051@shisu.edu.cn

**Abstract:** Pedagogical translanguaging has emerged as an important strategy facilitating the sustainable use of English as a Medium of Instruction (EMI) in educational settings. This mixed-method study, conducted in an EMI finance classroom at an international school in Shanghai, China, investigates the translanguaging practices of students in classroom interactions as well as their attitudes toward translanguaging as a communicative and pedagogical strategy. Drawing on video-assisted classroom observations and semistructured interviews, this study reveals that the participants' translanguaging practices are motivated by ease of communication, facilitated by contextual resources, and reflect their strategic maneuvering of the linguistic resources in their repertoires. The data also suggest that the participants are generally positive about translanguaging as an aid in comprehension and for the enhancement of content learning. Some participants, however, expressed reservations about the acceptance of translanguaging as a standard, formal linguistic choice. The findings suggest that EMI teachers should recognize the linguistic resources of students in their entirety and incorporate them into classroom activities to promote biliteracy and the learning of academic content.

**Keywords:** translanguaging; EMI; international school; pedagogy

## 1. Introduction

The use of English as a Medium of Instruction (henceforth EMI) has seen a rapid increase in scale across the globe [1]. EMI is defined as "the use of English language to teach academic subjects (other than English itself) in countries or jurisdictions where the first language of the majority of the population is not English" [1] (p. 19). Recent years have also witnessed a rise in the number of EMI studies conducted in higher education institutions across various academic disciplines [2], indicating that English is not only taught as a skill-based subject but also serves as an instructional language to assist in teaching other disciplines [3].

The effectiveness of bilingual education in China, with particular reference to academic discourse, has been widely researched, though not entirely with the focus on secondary education [4]. In China, Putonghua (the official national language) is employed to teach subjects in public (and partially private) primary and secondary schools. However, the situation is very different in international schools, where English functions as the main instructional language for a wide range of course content, including science, mathematics, history and literature, among others [5]. Although English is designated as the sole language for use on campus and in the classroom, it is not the only language used in EMI classrooms at international schools. Due to the linguistic resources shared by the teachers and students, teaching may be delivered in languages other than English. As a result, translanguaging often occurs during class time.

The concept of translanguaging, which originated in the Welsh bilingual context, has been associated with different terms such as code-meshing [6], metrolingualism [7], and the continua of biliteracy [8]. These terms have been introduced in multilingual settings to describe the flexible maneuvering of interlocutors between different language entities [9]. Translanguaging, instead of treating languages as separate entities, aims to consider the entire language repertoire of the speaker. This perspective emphasizes that the use of original and complex interrelated discursive practices can no longer be easily assigned to a traditional definition of a language [10]. Consequently, researchers have come to view translanguaging practices conducted in EMI classrooms from a new perspective; they have begun to interpret these practices as an effective way of meaning-making, a function that aligns with the need for content learning in EMI contexts.

Due to expanding interest in pedagogical translanguaging in educational contexts, studies have begun to explore translanguaging practices in EMI classrooms. While most research focuses on teachers' translanguaging practices in higher education institutions in China, a research gap remains related to the medium of instruction in primary and secondary classrooms. In particular, this applies to international school contexts where a rising number of bilingual or multilingual students are learning subjects via the medium of English. To understand how teachers and students in international schools promote learning through the flexible deployment of their multilingual linguistic repertoires, and to provide insights into students' learning processes, the current mixed-method research investigates students' translanguaging practices and attitudes, and analyzes their attitudes towards classroom linguistic practices. It is hoped that these insights will be of use to EMI teachers in both public and private institutions, encouraging them to actively utilize pedagogical translanguaging strategies to facilitate content learning.

## 2. Literature Review

### 2.1. Translanguaging

Translanguaging began as a pedagogical practice in Welsh/English bilingual education [11] which later spread across the globe as views on multilingualism changed in the late 20th century [12]. The theorization of translanguaging has been facilitated by abundant evidence of fluid and dynamic translanguaging practices. García [13] (p. 45) describes translanguaging as "multiple discursive practices in which bilinguals engage in order to make sense of their bilingual worlds." The term translanguaging differs from previous linguistic terms such as code-switching and code-meshing: it depicts a paradigm shift that challenges the existence of "languages" as identifiable and distinct systems [14].

Translanguaging accurately captures the complex and novel essence of the development of language and provides ample affordances that the Post-Multilingualism era demands [15]. What matters is no longer the number of languages an individual has at their disposal but the way they mobilize their entire linguistic repertoire, including their semiotic, modal, and sensory resources, to facilitate the meaning-making and delivery process [10,16]. Translanguaging introduces open discursive exchanges, enabling people to recognize their particular languaging values. This in turn allows fluid discourse to flow, unleashing the potential to reveal new social realities [17].

### 2.2. Pedagogical Translanguaging

Translanguaging has been explored in both naturalistic and educational settings [18–21]. While in naturalistic contexts emphasis is often placed on the flexible and creative employment of one's whole linguistic resources among bilingual communities in public domains [18,22], including in Internet spaces [23], translanguaging in educational settings poses different communicative challenges and opportunities. Attention is mainly paid to the exploration of the possibilities of enhancing comprehension and increasing learning effectiveness by actively mobilizing both the teachers' and students' full linguistic repertoires [24–26].

Translanguaging was first conceptualized as a pedagogical practice at the beginning of the 21st century [27] and, in recent years, has become a focus for researchers concerned with multilingual education [28,29]. Li [30] has argued that translanguaging provides a space for multilingual language speakers to combine different dimensions of their identity into one coordinated and significant performance, resulting in a more meaningful experience. This conclusion has laid the foundations for translanguaging practices across many situations, particularly in classroom settings. Reviewing studies on translanguaging as pedagogy, Conteh [16] argued that previous research tended to emphasize the process of interaction rather than its pedagogic potential. For this reason, more research is needed to more effectively explore pedagogical translanguaging.

Empirical evidence has revealed that translanguaging can be employed as a deliberate strategy to achieve a number of pedagogical outcomes, including explaining subject content, eliciting students' oral output, managing classroom discipline, and building teacher-student rapport [31–34]. Translanguaging has been used to describe not only multilingual oral interaction but also the inclusion of different languages in written texts [6,35]. More recent studies have also analyzed teachers' and students' attitudes to classroom translanguaging practices. Such studies aid in raising teachers' awareness of translanguaging as an effective pedagogy [36–38].

### 2.3. Pedagogical Translanguaging in EMI Classrooms

The pedagogical functions of translanguaging in primary, secondary, and tertiary CLIL science classrooms have been extensively researched to better understand its role in knowledge coconstruction [20,39–41]. These classrooms usually follow an agenda of bilingual teaching and learning and consider academic language acquisition to be as important as content learning [42,43]. A brief review of recent literature reveals that secondary science classrooms appear to be the most frequently selected setting for translanguaging studies in EMI contexts [44,45]. For instance, Tai and Li [46] discovered that translanguaging can be used to create a playful, informal atmosphere in EMI science classrooms in order to facilitate communication and content learning as well as promote interpersonal relationships. Additionally, Infante and Licona [47] describe how translanguaging in a bilingual middle school science classroom was used to clarify the meanings of professional terms, connect with students' previous knowledge of the topic, and engage students in various scientific practices (e.g., observing, questioning, hypothesizing, and explaining). These three functions formed a base for identifying the impact of teachers' translanguaging practices regarding content learning, and illuminating the purposes and reasons behind students' own translanguaging performances in EMI classrooms.

The most relevant studies related to translanguaging in EMI classrooms build their arguments from the perspective of colearning between teachers and students and the students' ability to access scientific knowledge. In the context of scientific content learning, Mazak and Herbas-Donoso [14] described translanguaging as "strategic, dynamic, and woven through the presentation of academic content," and as part of a trajectory that aims to help improve student proficiency in English for scientific purposes in higher education science classrooms (p. 699). Meanwhile, Tai and Li [48] examined how EMI teachers and students jointly gain new knowledge in bi/multilingual classrooms through translanguaging.

Bringing outside discourse and content into the classroom is another focus of translanguaging studies in EMI settings. Teachers are found to be skilled at mobilizing multilingual resources and engaging in multimodal approaches when incorporating knowledge acquired outside classrooms. This is believed to significantly enhance the contextualization and interpretation of complex scientific concepts [32,49].

The current research aims to contribute to the growing body of literature in translanguaging practices in EMI classrooms. This study differs from previous research in its emphasis on students' translanguaging practices and their subsequent feedback. Importantly, this approach allows teachers to reflect on their current practices and to consider



new ways in which to create a better classroom ecology. It is worth knowing the reasons behind students' translanguaging practices and attitudes to further understand how much their content learning and self-delivery are influenced by these practices. Findings generated from this research may influence course instructors to modify their pedagogical strategies accordingly. In addition, the current study adds to the limited body of research on translanguaging practices in EMI classrooms at international schools in China. Whereas most secondary schools in Chinese mainland are state-owned, the existence of private international schools is not uncommon, particularly in large municipal cities. International schools, in theory, only host students who do not possess passports of other forms of identifications from the Chinese mainland, and stipulate English as their official campus language. Courses are taught in English, and students are expected to communicate in English on campus. Nevertheless, given the fact that teachers and students are usually bilinguals of English and Putonghua, an intertwined use of both languages often occurs in classroom interactions. Unlike previous studies, which focused largely on secondary science classrooms, this study explores translanguaging in an EMI finance course to determine whether and how students exploit their entire linguistic repertoire to make meaning and construct knowledge in a supposedly all-English learning environment.

## 3. Methodology

The study explores the translanguaging practices that take place in an EMI classroom at an international school in Shanghai, along with the students' attitudes toward the use of translanguaging, both as a form of classroom discourse and a means of communication. It aims to interpret the dynamic and fluid classroom interactions in EMI classrooms from the translanguaging perspective. In addition, the subsequent analysis examines the extent to which one's entire linguistic repertoire can facilitate the learning of various subject contents. It is hypothesized that intermixing and flexible use of one's full linguistic resources, which could enhance the comprehension of subject knowledge, would be witnessed in classroom interactions, and that translanguaging as an effective communicative tool would be welcomed and accepted by the students. Hence the study aims to address two research questions:

What types of translanguaging practices among students can be identified in EMI finance classes at an international school in Shanghai?

What are the students' attitudes toward classroom translanguaging?

Two data collection methods were utilized to explore students' translanguaging practices and attitudes from both qualitative and quantitative angles. Video-assisted classroom observations were used to record classroom interactions. Course recordings effectively capture the students' maneuvering between multiple languages and provide an opportunity to observe the classroom community, as well as allowing the researcher "to document those processes in even greater detail and precision than is possible with ordinary participant observation and interviewing" [50] (p. 204). Unlike audio recordings, videos offer visual evidence of the emotional changes and physical expressions among students. In turn, this method helps researchers to collect firmer evidence of the challenges faced by students. Interviews, on the other hand, enable the researchers to gain in-depth knowledge of the reasons behind their decision-making processes. In this particular study, interviews made it possible to analyze the beliefs students hold in relation to the flexible deployment of their multilingual competence captured by the classroom video recordings. Interviews also "yield direct quotations from people about their experiences, opinions, feelings and knowledge" [51].

### 3.1. Context and Participants

This study was conducted at a private international school located in Shanghai, one of the most cosmopolitan cities on China's eastern coast. The EMI classroom features an annual short-term extra-curricular course that aims to cultivate students' awareness of money management from a young age and assist them in their choice of future studies. It

is organized in the form of one four-hour intensive lecture followed by six camp activities over three days. The camp activities last for four to five hours per day and take place either in classrooms or real-life outdoor settings where students sell goods at shopping malls, imitate the procedures of depositing and saving money in banks, and participate in quizzes that test their economic and financial knowledge. The in-class lecture and camp activities are divided into three themes: economics, finance, and entrepreneurship. The level of difficulty is adapted to the cognitive level of the students based on their age. English is the medium of instruction, and textbooks, teaching aids, and classroom signs are written in English. However, as both students and teachers are fluent in Putonghua, the language of instruction often involves a flexible and natural integration of Putonghua and English.

The participants in this study included three teachers and 40 students. The students were from Grade 5 to Grade 8 or within the age group of 12 to 15. Their English proficiency varied due to their different ethnic backgrounds, family languages and previous learning experiences, ranging from CEFR B1 to C1. Students of Chinese ethnic backgrounds in general possess a higher level of Putonghua, but all students were able to conduct daily conversations freely in Putonghua. All teachers were native speakers of Putonghua and proficient users of English. Jack and Isaac had 3 years of experience studying in the US, and Vicky acquired an IELTS certificate as a fluent English speaker (All teacher and student names in this article are pseudonyms). Therefore, these three teachers were qualified to fully use their linguistic repertoires and shift between Putonghua and English, employing translanguaging as a strategy to facilitate teaching.

The selection and sampling of participants were purposeful, criteria-based, and recursive [52]. Translanguaging requires language speakers to have multilingual competence, which many ethnic Chinese individuals possess due to their overseas experience and language environments. In general, English is the campus language for all international schools in China, and is the only authorized medium of instruction for all campus academic and extra-curricular activities. The international school in this study aims to provide a curriculum designed especially for its international students who reflect a diverse assemblage of cultural backgrounds: 34 of the aforementioned 40 student participants were from more than 10 countries including the US, the UK, Australia, New Zealand, Italy, France, etc., while the rest were from the HK, Macao, and Taiwan regions of China. Their multilingual competence was considered strong, which enabled them to employ translanguaging in class to make meaning.

### 3.2. Methods and Procedures of Data Collection

Consent from the participants was given and their confidentiality, anonymity, and voluntariness were guaranteed before data collection was initiated. The data corpus of this study comprises classroom observation and semistructured group interviews conducted during the courses taught between October 2020 and January 2021. Observational data were collected through video recordings and field notes to help generate an in-depth understanding of the linguistic and social practices of teachers and students [53]. A total of 75 teaching hours (the in-class lecture and camp activities) were observed and video-recorded.

This study also drew on three semistructured group interviews with students (two students in each group). Students were asked about their willingness to participate in group interviews via a mobile survey application. In the end, six students volunteered to join the interviews. Each interview lasted for 30 to 40 min, and the researchers used an interview guideline to elicit participants' views. To avoid potential ambiguity and misunderstandings caused by the inaccurate use of a less fluent language, participants were allowed to answer the questions in either Putonghua or English. The guiding questions asked for students' opinions about translanguaging pedagogy compared to a monolingual approach, the reasons for their preferences, and the influence of classroom translanguaging practices on their language proficiency and content learning.

*3.3. Data Analysis*

The process of data analysis was primarily inductive and recursive, developed in tandem with the data collected [54]. Categories and terms emerging from the data set, and recurring codes found during the data collection process, were further verified with the participants. Once the data collection was complete, connections between the various types of translanguaging practices were explored, studied meticulously and compared to those from previous research [14,31,34,47,48]. The analysis also adopted a cross-case analysis [51]; when the salient categories were recognized, they were compared to the data collected across other participants until they were fully revised and confirmed.

The first research question was answered mainly through data collected from video-assisted classroom observation, including video footage stored in computers and hand-written notes with detailed descriptions. Instances of classroom discourse that included the flexible shuttling between languages within one sentence were transcribed verbatim and studied in comparison with a list of coding protocols generated from previous studies until categories relevant to the current study were identified. Data from partial interviews with specific participants were also of assistance. Particular attention was paid to interactional microcontexts wherein participants made full use of their linguistic resources to enhance interaction. Additionally, results generated from interviewing data were used to answer the second research question. Content analysis, which is frequently used to analyze instructor interview data, reflective journals, teaching documents, and observation field notes at the macro level, were applied to interview transcripts in this study in order to understand student perceptions and attitudes toward classroom translanguaging practices. This is a broad term often used to "characterize the collection of generic qualitative analytical moves that are applied to establish patterns in the data" [55] (p. 245). Different from quantitative content analysis, qualitative categories in content analysis are not preconceived, but rather derived inductively from the data. This is pattern-finding as well as iterative [55]. Transcripts of interview recordings were subject to initial coding and second-level coding before core themes were conceptualized, compared, and validated [56]. Based on their salience and relevance to other important themes in the domain, a number of overarching themes were selected to be further interpreted and elaborated on. After the interviews, the researchers remained in contact with the participants, asking for further clarification of their interview statements when necessary.

## 4. Findings

The results generated from the analysis are presented in this section and organized using the two research questions of this study as headings.

*4.1. Student Translanguaging Practices in EMI Classrooms*

Overall, since the classroom discourse in the present study involved the intermixing of Putonghua and English, translanguaging practices tended to differ among participants depending on their first languages (L1s). When students spoke Putonghua as their L1, their translanguaging practices showed the dominant presence of Putonghua for their primary sentence structure, with English words interspersed; this pattern was reversed for students whose L1 was not Putonghua.

In contrast to previous studies, this study found that the translanguaging practices of students are influenced by a wide variety of factors. Overall, this study indicated that the three most noticeable elements triggering students' translanguaging practices were their ease of communication, contextual resources, and ability to strategically maneuver their linguistic resources in their repertoires, as demonstrated by instances extracted from class transcripts.

4.1.1. Translanguaging Practices Motivated by Ease of Communication

Such translanguaging practices were evident during the active thinking process when students employed discourse markers to fill in the pauses and gain additional time for

subsequent expressions, or simply for ease of communication, as they explained in the interviews that followed. The most common English filler word used for explanatory purposes was "like." Counterpart expressions in Putonghua such as "就是", "然后", "的" (the English equivalents of "that is," "and then," and "of") were also utilized frequently. Examples 1 to 3 illustrate translanguaging examples in this category.

EXAMPLE 1:

(TA: Teacher A)

TA: So, what is Business Plan, 我们所说的商业计划书呢?

*What is the Business Plan we talked about?* (Words in italics are English translations)

Stanley: 商业计划书 , it was like, 去做一个展示去吸引 . . . 哦!投资人!

*Business Plan, it was like, to make a presentation so as to appeal . . . oh, investors!*

In Example 1, in response to the teacher's question, the student interchangeably utilized Putonghua and English to explain a term listed in the textbook and a choice he had made. He also used the colloquial filler word "like" to introduce a definition. It was later confirmed that he was able to convey his meaning solely in one language. The reason why he spoke in this way, as he further explained, was that the filler word slipped out of his mouth and he subconsciously employed translanguaging to complete his sentences. He also mentioned that when encountering obstacles in conveying his meaning, he preferred to use English filler words because they bought him more time to think and boosted his verbal performance.

EXAMPLE 2:

(TB: Teacher B)

TB: Jolie, 你觉得王林为什么愿意买 Molly 这个 IP 呢?

*Why did Wang Lin wish to buy the IP, Molly?*

Jolie: 因为王林是 Popmart 的 boss, Molly will make him money

*Because he is the boss of Popmart, and Molly will make him money.*

TB: Then why can't others use Molly for money after Wang bought it?

Chloe: 因为 Molly 的 designer gave the copyright to Wang.

*Because Molly's designer gave the copyright to Wang.*

As is shown in Example 2, the most frequently employed discourse marker in Putonghua was "的", which serves as a genitive structure in English. In the flow of conversations, it could be seen that both students actively chose this linguistic structure for ease of communication. They appeared to prioritize the content they wished to express over the form of expressions. In such contexts, translanguaging functioned as a practical and convenient tool for meaning-making and the facilitation of communication.

EXAMPLE 3:

(TB: Teacher B)

TB: So, guys, do you want to be an entrepreneur in the future?

Joyce: Yes, I want to start my own company 就是像 Elsa sweet shop 一样, 然后 eat lots of sweets.

*Yes, I want to start my own company like Elsa sweet shop, and then eat lots of sweets.*

TB: A very good idea. How about Cindy?

Ian: I also want to run a business, but a bigger one, like . . . 就是像阿里巴巴那种, 然后 in this way I can become a trillionaire!

*I also want to run a business, but a bigger one, like... Alibaba, and then in this way I can become a trillionaire!*

Example 3 illustrates the use of two additional filler words in Putonghua. The phrase "就是" included in this exchange means "like" or "such as," while "然后" refers to "and then" or "afterward." In fact, the class transcripts revealed that 36 out of 40 students constantly used these two expressions: Ian said "就是" 32 times and "然后" 24 times in his 10-min-presentation. He confirmed that these two fillers often occurred in his monolingual Putonghua conversations as well, which bought him extra time to think about the content that would follow. When he spoke in English, the deployment of these discourse markers helped facilitate his delivery.

4.1.2. Translanguaging Practices Facilitated by Contextual Resources

Contextual resources in this study are defined as a range of elements that are believed to affect the participants' linguistic choices in classroom interactions, encompassing words and phrases in textbooks, PowerPoint slides, and signs in the classroom, as well as teachers' and/or peers' previous discourses. This type of translanguaging accounted for 62% of the total number of translanguaging instances recorded and observed in the classroom videos. Speaking mainly in English, students needed to refer to words that occurred in textbooks in Putonghua and were inclined to follow teachers' previous discourse patterns. Examples 4 to 6 provide evidence for translanguaging practices of this type.

EXAMPLE 4:

Sofia:　就像 aristocrats 信任 Columbus 一样, 如果你 trust 一个 company, 你就会投资他.

*If you trust a company, you will invest in it, just as aristocrats trust Columbus.*

In Example 4, the four English words Sofia used were presented in the textbook and the teaching aids, such as the PowerPoint slides, which she used as a basis for her presentation. As she explained later, in this case, translanguaging saved her the effort of translation, avoided possible pauses, and drew the attention of her peers to the slides. Here, her translanguaging practice was impacted by nonliving items in the classroom (i.e., textbook and slides) and proved efficient in facilitating her delivery of the presentation. It also demonstrated her acquisition of the content learned from the textbook.

EXAMPLE 5:

(TA: Teacher A)

TA:　So we can conclude that Higher Risk, Higher Return. Tiffany 你从这一页讲义中可以学到什么呢?

*What can you learn from this handout, Tiffany?*

Tiffany:　如果你想 higher return, 那就要面对 higher risk. 相应的, 低回报意味着低风险.

*If you want higher return, that means you will take higher risk. Accordingly, lower risk, lower return.*

In Example 5, a rule in the discipline of finance, "Higher Risk, Higher Return," did not occur in written form but was repeated as a key point by the teacher. When students took turns participating in an oral quiz, it was clear that after her learning in the course, Tiffany was able to paraphrase the financial rule by employing the strategy of translanguaging. She first borrowed the original English words from her teacher and then included them in an elaboration of the rule in order to respond to the teacher's question. This demonstrates that Tiffany was capable of using her full linguistic resources to indicate that she had understood the course content in both languages and that her bi-academic literacy had been enhanced by the teacher's classroom discourse.

EXAMPLE 6:

(Tiffany shouted to another student James, pointing at the sign):

Tiffany:　不要乱扔糖纸, Classroom Hygiene, you see?

*Don't throw the trash about! Classroom Hygiene there, you see?*

Example 6 showed how signs in classrooms could affect student's language choices. "Classroom Hygiene" was a sign posted on the wall of the classroom to encourage tidiness. On this occasion, Tiffany cited the sign and built it into a more convincing argument, as the rhetoric in Putonghua began with an imperative voice, thus creating a stronger effect. Tiffany's use of translanguaging in this microcontext was clearly affected by the nonliving item in the classroom and also reinforced the tone of her speech.

4.1.3. Translanguaging Practices Reflecting Strategic Maneuvering of Linguistic Resources in Their Repertoires

All participants in this study could be considered fluent bilingual speakers of Putonghua and English, though some might have possessed a stronger proficiency in one language or the other. From a translanguaging perspective, their flexible and intertwined

use of a fluid combination of Putonghua and English was not in any way evidence of their limited competence in either of the languages but rather proof of their ability to mobilize different linguistic resources, as can be viewed in Example 7 and 8.

EXAMPLE 7:

James:　As we all know, Pony Ma 是 Tencent 的发明人.

*As we all know, Pony Ma is the inventor of Tencent.*

Ian:　　No, he is 创始人!

*No, he is the " founder "!*

EXAMPLE 8:

Queena:　马化腾觉得 . . . [pause] beeper inconvenient, 所以对 beeper 进行改造, 最后发明 OICQ.

*Pony Ma thought the beeper was inconvenient, so he renovated it and finally invented the software OICQ.*

Applying the traditional monolingual lens to these examples could result in the conclusion that the students shifted between two languages because they were incompetent in both. However, the researchers believe that a translanguaging analytical approach offers a more convincing and updated explanation for their bilingual strategy. Example 7 was captured when James was giving a presentation in Putonghua assisted by slides in English which displayed the words "Pony Ma," "Tencent," and "founder." Both students might have initially learned the concepts of "inventor" and "founder" in Putonghua and therefore chosen to maneuver between different linguistic resources to facilitate the interaction. A similar microcontext can be seen in Example 8, where Queena, probably unfamiliar with the counterpart of the word "beeper" in Putonghua, decided to draw on her whole linguistic repertoire to make herself understood in the presentation. Both examples indicate that in an EMI setting, where all interlocutors are bilingual/multilingual, the deployment of translanguaging is conducive to maintaining the conversational flow of interactions and enhancing comprehension, hence contributing to a more sustainable classroom ecology.

*4.2. Student Attitudes to Classroom Translanguaging Practices*

Student attitudes to classroom translanguaging practices were analyzed via the results gained from group interviews. Six participants were selected for group interviews after offering their consent and willingness to cooperate in the study. Five of the six participants involved in the group interviews generally favored translanguaging, while the remaining participant supported the monolingual approach. Interviewees expressed a variety of attitudes towards translanguaging practices in EMI classrooms in the semistructured group interviews. Four of them clearly preferred the translanguaging approach to monolingual alternatives to support content learning and fluent delivery, as exemplified in Joey's answer:

*"I fancy the method we employed in the past few days. Sometimes I would suddenly forget some words in both languages, and as I'm allowed to interchange, I can speak more fluently. At school, I can only speak English in class, because some teachers cannot speak Putonghua. So sometimes I cannot make myself understood."*

According to Joey, the monolingual approach sometimes interferes with his fluent delivery and process of meaning-making, and translanguaging practices helped to promote his ease of communication in class. Another common phenomenon was also revealed at international schools in Shanghai in that some teachers are unfamiliar with the official language or the local dialect. However, students might access some knowledge in other languages. This disconnect could potentially hinder the teaching and learning of terminologies in subjects such as mathematics and physics, therefore decreasing students' learning efficiency.

One participant, Gordon, agreed with the positive effects of translanguaging, but also expressed his concerns about the drawbacks of the strategy. He was an active translanguager both in class and in daily life, but embarrassing moments occurred, as he notes:



> *"It's really hard to say. I did benefit a lot at ME CAMP (the EMI course observed), but when I did my homework after school, I hesitated if it's okay to write like that. It's kinda inadequate."*

Gordon was not the only one who was concerned about the possibly informal nature of translanguaging and the limited occasions that were suitable for this type of language use. As Queenie points out:

> *"Isn't this way of speaking weird and informal? Nobody makes an important speech or announcement like this. I'd say it randomly erases the boundaries between languages. It will end up like an ugly mishmash and hinder people from seriously learning different languages."*

Gordon and Queenie's comments revealed a common concern about social acceptability and traditional perceptions of languages. As EMI courses primarily exist to convey knowledge of specific subjects and integrate students into the relevant discipline, the improvement of language ability is a secondary consideration. The main goals are to facilitate the process of student meaning-making and to help them understand complex terms that can only be interpreted with difficulty through a monolingual approach. However, while it enriches their bilingual academic repertoire, one possible drawback of translanguaging could be that students repeat their teachers' words in another language without thoroughly understanding them. Considering this pitfall, it would be advisable for teachers to confirm students' comprehension by requiring them to translate or paraphrase.

Moreover, as Queenie claims, on formal occasions translanguaging is less acceptable and regarded as careless or lacking in seriousness. This may be a result of public stereotypes about the definition and function of languages, as well as the local trend of selecting English as a perceived "superior" medium of instruction. The two students' assertions indicate a "pecking order" [57] (p. 377) between the mediums of instruction that is deeply ingrained in the subconscious assumptions of Chinese students and parents. As one of the mediums of instruction, English is perceived as the dominant language, somehow more prestigious than Putonghua, especially among international schools in China. Yet the fact is, when it is allowed in classrooms, translanguaging can enhance the efficiency of content learning and meaning-making, as well as improve the mobilization of students' entire linguistic repertoires.

## 5. Discussion

This study, conducted in an EMI finance course, investigated translanguaging practices at an international school in Shanghai, China. Particular attention was paid to two main issues: students' translanguaging practices, including their underlying motivations, and students' attitudes toward translanguaging as a form of classroom discourse. Through the use of video-assisted classroom observation and group interviews, the researchers found that students' translanguaging practices can be categorized into three types: (1) translanguaging practices motivated by ease of communication; (2) those facilitated by contextual resources; and (3) those reflecting students' strategic maneuvering of linguistic resources in their repertoires. Furthermore, most of the student participants expressed a preference for an intermixing of Putonghua and English as a means of communication in class, rather than the traditional monolingual approach to teaching. It is believed that translanguaging as a form of classroom discourse helps to develop students' bilingual academic knowledge, reinforcing their learning of the subject content and strengthening their ability to make meaning by drawing on their entire linguistic repertoire.

Compared to previous studies, results from the current research showed a certain degree of diversity in relation to its research questions. As for translanguaging practices, findings from this study were generated from the perspective of factors that have triggered the deployment of translanguaging, rather than the functions of such practices in completing pedagogical tasks, as often demonstrated by previous research [32,33,41,42]. Nevertheless, Mazak and Herbas-Donoso [14]'s viewpoint of translanguaging being prompted

by contextual resources is echoed in this study. In terms of students' attitudes to classroom translanguaging, results from the present study are in line with some of the existing literature [5,34] but contradict with Wang [33]'s findings, where more than half of the participants opted for a monolingual approach of teaching.

Findings from the present study contribute to the understanding of translanguaging practices in EMI classrooms from two perspectives. Firstly, the study was contextualized in an EMI finance course at an international school on the Chinese mainland, where little relevant empirical evidence has hitherto been produced. Previous research has tended to focus on classrooms where a more multilingual tradition prevails, or where English used to be, or still is, one of the official languages [32,44,45,48]. Nevertheless, although English is the designated campus language for almost all international schools in the metropolitan Shanghai area, these schools also function as ideal environments for interlocutors to incorporate various linguistic resources and engage in meaningful multilingual conversations to enhance content learning and improve biliteracy. The diverse linguistic backgrounds of the teachers and students in these schools provide significant potential for future studies in translanguaging.

Secondly, most previous research exploring translanguaging practices in bilingual science classrooms has focused on the teacher as the main translanguager. Through analyses of classroom discourse among teachers, different translanguaging strategies emerged [31,34,47] and specific pedagogical purposes such as bridging outside knowledge and colearning were highlighted [14,49]. Even the limited number of studies that did prioritize student contributions in classroom interactions showed more interest in categorizing students' translanguaging practices according to classroom activities [41,58]. The present study aimed to understand the translanguaging practices of students from the viewpoint of their motivations instead of emphasizing pedagogical functions in micro-teaching contexts. This perspective has given the researchers a fresh perspective and allowed for an in-depth approach to interpreting the internal and external factors that affect student language choices in class.

Furthermore, findings from this study also concurred with what Li [30] termed translanguaging space. This is a space where different social practices and linguistic codes are converged and interacting with each other, which connects not only language users and their linguistic repertoires but also their experiences, attitudes and ideologies into an orchestrated performance [15]. In this study, regardless of the students' knowledge of English as the medium of instruction, and their impression of translanguaging as informal, they were still very willing to create a translanguaging space where they enjoy the freedom to incorporate their entire linguistic repertoire in communication. It can be argued that such translanguaging space fosters a healthy and sustainable teacher-student relationship, as it moves the dynamic between them away from the traditional hierarchy to become a harmonious and mutually improving ecology. Together, they break the monopoly of English as the only language in EMI finance courses. During this process, students develop a positive feeling about themselves as fluent bilinguals and the effectiveness of their content learning is increased.

From a pedagogical viewpoint, the present study examined students' mobilization of multiple linguistic resources, as they interchangeably made full use of their prior knowledge and learning experiences to master the subject knowledge at hand. They did not limit themselves to any one particular language: on the contrary, they actively deployed translanguaging as a valuable communicative tool. These findings raise crucial pedagogical implications for teaching EMI courses. On one hand, teachers are expected to recognize students' linguistic resources instead of viewing them as an impediment to the content learning process. In fact, teachers are encouraged to incorporate these linguistic resources into classroom activities to promote the acquisition of biliteracy and academic content. On the other hand, as Poza [41] warned, teachers should be wary of oversimplifying translanguaging as a pedagogy. Students should be guaranteed authentic input in the target forms if they are among the learning outcomes. It is also of vital importance to have access to

students' attitudes to teachers' translanguaging practices. As evidenced in this study, some students expressed wariness when it came to the legitimacy of using translanguaging in class. Teachers are suggested to obtain a full picture of students' views on translanguaging as a pedagogy so as to adjust their classroom linguistic choices accordingly.

## 6. Conclusions

In summary, this study revealed three types of translanguaging practices: those motivated by ease of communication, facilitated by contextual resources, and reflecting their strategic maneuvering of the linguistic resources in their repertoires. It was also discovered that the participants in general held positive attitudes to translanguaging as an effective tool in enhancing understanding and the learning of subject content, despite reservations expressed by some participants concerning its acceptability as a standard linguistic code.

The current study illuminates students' motivations behind and attitudes toward their translanguaging practices; based on these insights, pedagogical suggestions can be developed and refined. This research adds value to the exploration of translanguaging practices among students. These practices involve students' fluid and fluent use of their full linguistic repertoire to facilitate meaning-making, enhance classroom interaction, and demonstrate comprehension. The linguistic practices and preferences of students are of crucial importance for further research in pedagogical translanguaging, as they reflect how well students have understood and digested the subject content, which is important in creating a mutually beneficial and sustainable classroom ecology. This study is obviously not without limitations. A larger participant sample and a multimodal analysis of the recorded classroom videos would help to reveal further details surrounding students' translanguaging practices. Future research could continue to investigate translanguaging practices in EMI classrooms with a variety of course types and student age groups to examine how translanguaging can best foster EMI classroom sustainability.

**Author Contributions:** Conceptualization, X.Z. and X.G.; data collection and analysis, X.Z. and C.L.; writing—original draft, X.Z. and C.L.; writing—review & editing, X.G. All authors have read and agreed to the published version of the manuscript.

**Funding:** This work was supported by the Shanghai Municipal Education Commission (C2021236), Shanghai International Studies University (RG203509) and (2019114018), and Hermia Educational Consultancy (RG204128).

**Institutional Review Board Statement:** Not applicable.

**Informed Consent Statement:** Informed consent was obtained from all subjects involved in the study.

**Data Availability Statement:** The data presented in this study are available on request from the corresponding author.

**Conflicts of Interest:** The authors declare no conflict of interest.

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
