# Peer review of "Towards a Sustainable Classroom Ecology: Translanguaging in English as a Medium of Instruction (EMI) in a Finance Course at an International School in Shanghai"

_sustainability, doi:10.3390/su131910719_

Round 1

Reviewer 1 Report

  • Very informative introduction setting the background and identifying the research gap and importance of the study.
  • The literature review is relevant, informative and comprehensive with academic and up-to-date sources. Yet, it can be enriched regarding specific aspects mentioned in the article as notes.
  • Research questions are clear and there are some solid methodological justifications. Yet, this section misses on some important aspects (e.g. justifications on the use of the questionnaire are not enough and sources are missing; Content analysis is not explained as a theory. Check the comments in the article). Also, there are no hypotheses mentioned.
  • The findings are presented well with examples. But there are no grass for the descriptive analysis of the questionnaire. There are only percentages. Graphs would help.
  • The discussion section does not compare and contrast the findings of the study with findings from previous studies addressing the 2 research questions and the findings. Although there is some critical engagement, still this is inadequate.
  • In the conclusion, some implications and limitations are presented but clearer conclusions regarding the overall results of the study must be offered.
  • Overall, this is a very interesting and original research, but certain important aspects of the article must be improved and enriched.

Reviewer 2 Report

CLASSROOM ECOLOGY, TRANSLANGUAGING IN EMI

Very interesting article, changes could improve some gaps or clarify researchers´ points of view.

Line 11.  EMI is usually in University contexts, is this the case here? As the authors mention it as “ in an EMI finance classroom at an international school in Shanghai”, in schools is called CLIL. (further up in line 58, 112-113)

Lines 41/ 42. Implying translanguaging is more than two languages, check references  (line 72 they mention two languages).

Line 61.  The study is carried out in a private institution, but hopefully the effect or the learning will be for a much broader community (rewrite).

Line 174, which languages?

Line 189, specify that is a private institution and range of age of participants, English level…

Lines 204, authors specify type of teachers, all of them fulfil the requirement of those two languages, are they fluent in any other? And the students?

Line 217, …schools don´t admit students from China mainland… so all students, in theory, are foreigners, what will be their level of Putonghua? Do they master it?

Line 231, the questionnaire has two questions; maybe it should be called something else as to the type of format. Then it helped to select potential future participants, based on which aspects. Why select the participants, if maybe information could be gathered from all of them (contrasting).

Line 251, …contrasted with previous research….  A research done by the authors before or from other researchers?

Line 281… when students spoke Putonghua as their L1…. Explain, as line 217 doesn’t match with this appreciation

Line 462 TRANSLANGUAGER (any other similar terms?). His doubts about the positive/negative aspect of translanguaging are interesting, maybe expand on it giving references.

Line 539… translanguaging space… interesting term, but true translanguaging is limited to be used in a certaing context and with certain people (interlanguaging), give references related to the “space” concept.

Thank you

Reviewer 3 Report

Thanks for submitting this manuscript. It's an interesting topic about translanguaging and EMI. However, I do have some questions about the paper.

  1. As the paper focus on EMI in China, I think more relevant studies could be involved in your literature review. For example: Tong, F., Wang, Z., Min, Y., & Tang, S. (2020). A Systematic Literature Synthesis of 19 Years of Bilingual Education in Chinese Higher Education: Where Does the Academic Discourse Stand?. SAGE Open10(2), 2158244020926510.
  2. Regarding the instrument used to collect data, for example, the questionnaire, I think it would be better if the author could provide a more specific introduction of the questionnaire, including reliability and language of the questionnaire.
  3. In addition to classroom observation, do you use any coding protocol to analyze the videos? Or the observer simply make notes of the videos?

Round 2

Reviewer 1 Report

It is definitely a very improved version with important additions, deletions and revisions. I suggest that two hypotheses should be added to be aligned with the 2 research questions and present an even more complete idea of the study. However, the revised article can stand as it is. 

Once again, a very interesting topic to explore.

Author Response

Dear Reviewer 

Thanks very much for your comment. We have made some changes accordingly by rewriting the wording of the hypotheses so that they are better aligned with the research questions. Please refer to the first paragraph in the Methodology chapter. 

Many thanks 

Xiaozhou (Emily) Zhou, Chenke Li and Xuesong (Andy) Gao